# Microwave Irradiation: Effects on the Change of Colour Characteristics and Main Phenolic Compounds of Cabernet Gernischt Dry Red Wine during Storage

**DOI:** 10.3390/foods11121778

**Published:** 2022-06-16

**Authors:** Jiang-Feng Yuan, Yu-Ting Lai, Zhuo-Yao Chen, Hui-Xia Song, Jing Zhang, Da-Hong Wang, Ming-Gui Gong, Jian-Rui Sun

**Affiliations:** College of Food and Bioengineering, Henan University of Science and Technology, Luoyang 471023, China; yutinglai2020@163.com (Y.-T.L.); czychenzhuoyao@163.com (Z.-Y.C.); huixiasong@163.com (H.-X.S.); jingzhang@163.com (J.Z.); wangdahong2003@163.com (D.-H.W.); gongminggui@163.com (M.-G.G.); dasheng@haust.edu.cn (J.-R.S.)

**Keywords:** microwave irradiation, red wine, changing trend, colour properties, phenolic compound

## Abstract

Wine colour is an essential organoleptic property considered by consumers. In this paper, the potential effects on colour characteristics and the content of main phenolic compounds in red wine under microwave irradiation were investigated during wine storage. The results showed that the changing trend of colour characteristics of microwave-treated and untreated wines was very similar. Moreover, total phenolic compounds, total monomeric anthocyanins, main anthocyanins, main flavonoids, and main phenolic acids (gallic acid; caffeic acid; syringic acid; (+)-catechin; Cy-3-glu; Mv-3-glu) also showed similar change trends during storage. In other words, microwave irradiation had a long-term effect on the colour properties and main phenolic compounds of red wine, changes that require long-time aging in traditional processing. In terms of the studied parameters, the changes in microwave-treated wine were faster than those in untreated wine. These results showed that microwave technology, as a promising artificial aging technology, could in a short time produce red wine of similar quality to traditional aging.

## 1. Introduction 

Wine consumption in many countries, especially in China, has become an integral part of the social world [1]. Consumers’ demand for high-quality red wine has been increasing. To meet the requirements of consumers, producers need to produce high-quality red wine for the consumer market. Young wines have good visuals and rich fruit aroma, but young red wines can only be kept for a short time, so consumers prefer to buy aged wine. Aged wine can not only keep desirable sensory properties for a long time, but can also form new flavor substances after long-term aging [2]. In general, during aging complex chemical reactions occur between phenolic compounds, sugars, and acids, to reduce the bitterness and astringency of wines. In addition, the aromatic composition, colour, stability, and clarification of wines are further improved after this period of storage [3]. However, extended wine storage time does not improve wine quality; the taste and quality of wine decrease due to oxidation and browning when past the best time for drinking.

Wine aging has been a vital process in producing high-quality wine, and storage in oak barrels was the classic and efficient way. However, there are some shortcomings to the barrel aging process, such as the high cost of oak barrels, long aging time, and ample space for oak barrels, which had a severe impact on the production capacity and economic benefits of the enterprise [4]. Therefore, with the aim of ensuring the quality of wine, certain artificial aging technologies have been developed, such as oak products, micro-oxygenation, high hydrostatic pressure, magnetic field, irradiation, and others [5]. These technologies could reduce the aging time from a few years to a few months, improve wine quality, and reduce production costs. In recent years, microwave irradiation has become a popular research field in accelerating the wine aging process, because of its relatively low cost, high efficiency, shorter processing time, and pasteurization effect [6,7]. In addition, the high-frequency electromagnetic waves destroy the stabilization of weak hydrogen bonds by strengthening the rotation of polar molecules [8]. Free radicals are induced under microwave irradiation [9]; the existence of free radicals leads to complex chemical reactions in the aging process. Microwave treatment could reduce the use of SO_2_ [10] and improve sensory qualities of red wine [11]. 

Wine colour is an essential organoleptic property considered by consumers, which affects the commodity value of red wine [12]. During aging and storage, the unstable colour of young wine gradually transformed into stable and deeper colour, enhanced the 30–50% colour intensity of young wine, and changed from bright red to reddish-brown [13]. Anthocyanins directly imparted the wine colour according to their colour, and by the formation of more stable pigments through anthocyanin polymerization during storage and aging, namely, direct polymerization between anthocyanins and flavanols, or indirect polymerization between anthocyanins and flavanols via acetaldehyde [14]. Therefore, polymeric pigments are considered essential compounds in aged wines. Contrary to the observed polymeric pigments, the anthocyanin content decreased significantly during aging. Phenolic compounds, which were normally colourless, were the main chemicals that affect colour, astringency, and bitterness of red wine, and could improve the colour stabilization of aged wines by copigmentation reactions, through non-covalent interactions between anthocyanins and copigments to increase the stability of anthocyanins [15]. Therefore, the wine colour stabilization increased due to the copigmentation effect involving phenolic compounds during aging. 

Our laboratory conducted a preliminary studies on the change of the physicochemical characteristics of wine after a short period of microwave treatment [16], content change and degradation phenomena of phenolic compounds caused by microwaves [17], 1-hydroxyethyl free radicals induced by microwave [9], and the change of properties and activities of PPO effected by microwaves in the grape maceration stage [18]. As far as we know, the dynamic changes in the colour, main anthocyanins, and phenolic compounds in red wine under microwave irradiation during storage and aging have not been reported. In this study, the colour characteristics, main anthocyanins, and phenolic compounds were monitored after microwave treatment during the storage of young red wine. The study aimed to prove the positive effects of microwaves on changing wine colour properties, to determine further whether the colour-change trend of microwave-treated wine was similar to that of untreated wine during storage, and to identify the mechanism of the change of red wine colour properties caused by microwaves. 

## 2. Materials and Methods

### 2.1. Materials and Reagents

Cabernet Gernischt dry red wine of the vintage 2020 (alcohol level of 12.0%) was purchased from Yantai Weilong Winery (Shandong, China). Gallic acid (98.0%), caffeic acid (98.0%), (+)-catechin (98.0%), syringic acid (98.0%), cyanidin-3-*O*-glucoside (Cy-3-glu, 98.0%), and malvidin-3-*O*-glucoside (Mv-3-glu, 95.0%) were purchased from Hefei Bomei Biotechnology Co., Ltd. (Hefei, China). Before use, all standards were dissolved with methanol to 1 mg/mL and stored at 4 °C away from light. Deionized water from a Millipore filter system was used throughout the experiment (Millipore Co., Billerica, MA, USA). Methanol (HPLC grade) was purchased from Thermo Fisher Scientific (Madison, WI, USA). Folin-Ciocalteu reagent and formic acid were purchased from Tianjin Komiou Chemical Reagent Co., Ltd. (Tianjin, China). All other organic reagents and solvents used in the study were analytical grade. 

### 2.2. Microwave System

The microwave reactor was an XH-MC-1 microwave synthesis (2450 MHz, Xiang Hu Science and Technology Development Co., Ltd., Beijing, China), a low-temperature coolant circulating pump (Gongyi Yuhua Instrument Co., Ltd., Gongyi, China) provided −20 °C coolant, and the following experiments used a closed magnetic stirrer and controlled temperature cooling microwave irradiation system (CMCC-MI system) [17].

### 2.3. Preparation of Wine Samples and Storage

The colour changes of the young red wine were studied under different microwave conditions. Firstly, the effects of microwave power (100, 200, and 300 W) at 40 °C for 3 min were studied. Subsequently, the effects of microwave temperature (30, 40, and 50 °C) at 100 W for 3 min were studied. The untreated (M_0_) and microwave-treated samples were placed in a Schott bottle with screw caps, using three bottles in parallel. The wines were stored in the dark at room temperature (25 °C). Unless otherwise stated, the screw cap of the sample bottle was only opened to the atmosphere during measurement. Then, on the first, 10th, 20th, 30th, 40th, 50th, 60th, and 70th days after microwave treatment, a certain volume of red wine was taken from each for the determination of colour parameters.

In order to analyze the influence of microwave on red wine colour and main phenolic compounds during storage, the microwave treatment time and storage time of red wine were studied. The young red wines were treated with microwaves at 100 W, 40 ± 1 °C for 3 min and 6 min, respectively. The untreated wine (M_0_), the red wine treated for 3 min (M_t3_), and the red wine treated for 6 min (M_t6_) were each placed in Schott bottles (250 mL) with screw caps, and the three bottles were kept in parallel. M_0_, M_t3_, and M_t6_ red wine samples were taken every ten days to determine colour parameters and main phenolic compounds.

### 2.4. Colour Analysis

To evaluate the effects of microwave power and temperature on red wine colour, the colorimetric measurements were assessed by the CIELab space, using the CIE D65/10° illuminant/observer condition according to the OIV method. CIELab coordinates were determined by SC-80C automatic colorimeter (Beijing Kangguang Instrument Co., Ltd., China), including lightness (*L**), red/green colour coordinates (*a**), and yellow/blue colour coordinates (*b**). The *L** axis represented the wine lightness scale. The *a** value represented the degree of redness and greenness; the higher the *a** value, the redder the wine. The *b** value represented the degree of yellowness and blueness; the higher the *b** value, the more yellowish the wine. Colour difference (Δ*E***ab*), chroma (*C***ab*), and hue angle (*hab*) were calculated according to *L**, *a**, and *b** values of the tested wine samples [19]. The formula was as follows:(1)$$\Delta E^{*}=\sqrt{(\Delta L^{*2}+\Delta a^{*2}+\Delta b^{*2}})$$
(2)$$C^{*}ab=\sqrt{(a^{*2}+b^{*2}})$$
(3)$$hab=tan^{-1}\left(\frac{b^{*}}{a^{*}}\right)$$

The differences in each colour coordinate value between microwave-treated and untreated wine were shown with Δ*L**, Δ*a**, and Δ*b**.

A UV-2600 ultraviolet-visible spectrophotometer (Shimadzu, Kyoto, Japan) was used for scanning the spectrum between 380 nm and 800 nm, with a quartz cell of 1 cm path length. All the samples were diluted ten times with 12% (*v*/*v*) ethanol, and colour properties were determined by measuring the absorbance at 420, 520, and 620 nm, respectively. The sum of the absorbance at 420, 520, and 620 nm was the total colour intensity (CI).

### 2.5. Determination of Total Phenolic Compounds (TPC)

The determination of TPC was performed with Folin-Ciocalteu colorimetry [16], 15 μL red wine sample was added to the solution. Gallic acid equivalent per liter of red wine represented the result.

### 2.6. Determination of Total Monomeric Anthocyanins (TMA)

The content of TMA in red wine was determined according to the literature [16]. A sample of 1 mL 5-fold diluted red wine was added to the solution. The result indicated Cy-3-glu equivalents (mg/L).

### 2.7. Determination of Main Phenolic Compounds

HPLC analysis was carried out by ZORBAX SB-C18 column (5 μm, 4.6 mm × 250 mm, Agilent, Palo Alto, CA, USA) using an Elite 3100 Series HPLC system with a UV3100 detector. Before use, all mobile phases were degassed by ultrasound for 25 min and filtered using 0.45 μm membranes. Flow rate was 1.0 mL/min, column temperature was 25 °C, and injection volume was 20 μL. The parameters of mobile phases of major phenolic compounds referenced the literature [1].

Each sample was injected three times, and the chromatograms were recorded at 280 nm by the UV3100 detector. According to the retention time and ultraviolet–visible spectra of the corresponding standard, the phenolic compounds in red wine samples were identified at 280 nm, including gallic acid, caffeic acid, (+)-catechin, syringic acid, Cy-3-glu, and Mv-3-glu, then the concentrations of identified phenolic compounds were calculated by the calibration curve of each standard. The calibration curves were constructed by injection of five different concentrations of the standards, respectively.

### 2.8. Statistical Analysis

All the data were expressed as means ± standard deviation (SD) of three replications. Statistical analysis was performed by one-way analysis of variance (ANOVA) using statistical software SPSS version 23.0. Waller–Duncan multiple contrast testing (*p* < 0.05) was used to study the statistical differences in red wine colour properties during storage.

Principal component analysis (PCA) carried out with Origin version 2018was employed to reveal latent variables or factors that could explain the correlation pattern between the colour characteristics and the content of phenolic compounds in red wine.

## 3. Results and Discussion

### 3.1. Selection of Power and Temperature of Microwave

In order to determine suitable microwave power and temperature throughout the experiment, red wine was treated with different microwave power (100 W, 200 W, and 300 W) and temperature (30 °C, 40 °C, and 50 °C) to analyze its colour changes. CIELab parameters can more precisely define the chromatic properties of wine.

*L** values represented clarity from black (0) to colourless (100). As shown in Figure 1A, the *L** values of red wine treated with microwave irradiation from 100 W to 300 W were higher than that of M_0_ during storage. As the storage time extended from 1 d (the treatment day) to 70 d, *L** values of all red wine samples showed an apparent decreasing trend, which may be due to the combination of free anthocyanins and tannins forming more stable tannin–anthocyanin complexes and precipitation of pigment polymers; *L** values decreased with the extension of storage time [20]. It is well known that consumers prefer higher clarity of red wine; *L** values of microwave-treated red wine at different power were higher, which might have accelerated the chemical reactions of free anthocyanins with tannins under microwave irradiation, so the clarity of treated wines was higher than that of M_0_ during storage. *L** values declined by approximately 20 units from 1 d to 70 d in all wine samples, and the experimental results showed that different microwave power treatments did not change the trend and variation range of *L** values in red wine. However, red wine treated with different microwave power had higher clarity than M_0_, and red wine treated with 100 W had the highest clarity. As shown in Figure 1B, *L** values of red wine treated with microwave irradiation from 30 °C to 50 °C were higher than those of M_0_ at 1 d, *L** values of all red wine samples showed a decreasing trend with the extension of storage time from 1 d to 70 d, and *L** values of red wine with microwave treatment at 40 °C and 50 °C were higher than those of M_0_. The experimental results showed that microwave irradiation under different temperatures did not affect the change trend of red wine clarity. *L** values of red wine treated at 30 °C showed minor differences compared with M_0_, and red wine treated at 40 °C and 50 °C had higher clarity.

*a** represented red and green, which was an indicator to measure the contribution of anthocyanins to the redness of wine [21]. Due to anthocyanin oxidation reactions, *a** in red wine gradually decreased with aging. From the results shown in Figure 1C, with the extension of storage time from 1 d to 70 d, *a** of M_0_ showed a decreasing trend, indicating that the decrease of *a** may be due to the decline of anthocyanin content during red wine aging. Meanwhile, *a** of red wine treated with microwaves at 100 W, 200 W, and 300 W obviously decreased during the initial treatment process, that is to say, microwave treatment reduced anthocyanin content, resulting in a rapid decrease of *a** in a short time; *a** then showed an increasing trend at 10 d, this phenomenon showing that the *a** of red wine treated by microwave irradiation had a “regenerative” effect. *a** decreased gradually from 10 d to 70 d, but was obviously higher than that of M_0_ at 70 d, indicating that microwave-treated red wine had more appreciable redness at the later stage, and red wine treated with 100 W had the highest redness. As can be seen from Figure 1D, with the extension of storage time from 1 d to 70 d, *a** of untreated wine and microwave-treated wine at 30 °C showed a similar declining trend, which might indicate that microwave treatment at low temperatures could not cause significant changes in wine redness. After microwave treatment at 40 °C and 50 °C, *a** decreased significantly in the initial stage compared with M_0_, which might be due to the complex chemical reactions of red wine at higher temperatures, resulting in the decrease of anthocyanin content, so *a** decreased significantly. *a** increased on the 10th day, then gradually dropped from 10 d to 70 d, which may be because the physicochemical properties of red wine impacted by microwaves were not sufficiently stable, resulting in the reformation of anthocyanins. Red wine with microwave treatment at 40 °C and 50 °C had higher *a** values at the later stage, indicating that microwave irradiation at 40 °C and 50 °C could not only accelerate wine aging, but also produce red wine with more pleasing redness. Therefore, according to the effect of microwave treatment on *a**, microwave irradiation at 40 °C or 50 °C with 100 W could increase the redness of red wine.

*b** represented yellow and blue. Carvalho [22] reported that *b** of sweet, medium sweet, and dry wine showed an upward trend in the vinification and the initial stage of wine aging, then showed a downward trend from the second month to the 24th month. From the results in Figure 1E, with the extension of storage time from 1 d to 70 d, *b** of M_0_ showed a decreasing trend. The *b* * of microwave-treated red wine at 100 W, 200 W, and 300 W had an increasing trend from 1 d to 10 d, then a gradually decreasing trend from 10 d to 70 d. The *b** of red wine treated by microwave was lower than that of M_0_ at 1 d, indicating that microwave irradiation had the function of rapid aging. In the storage stage, *b** change was low, indicating that *b** was stable after microwave treatment; the change rate of *b** in M_0_ was 73.97%, and the change rate of *b** at 100 W was 13.66%, indicating that the yellowness of red wine after 100 W tended to be more stable, and the red wine tended to aged. From the results shown in Figure 1F, with the extension of storage time from 1 d to 70 d, *b** of M_0_ and red wine treated at 30 °C showed a similar decreasing trend, and *b** of red wine treated at 40 °C or 50 °C showed a slow downward trend, indicating that the yellowness of red wine after higher temperature treatment tended to be more stable, and the red wine tended to age.

Total colour difference (Δ*E***ab*) considered all differences between *L**, *a**, and *b** of red wine, and was used to evaluate the relationship between numerical analysis and visual perception [23]. In the colour evaluation of the wine, the colour difference (Δ*E***ab*) between treated red wines and M_0_ could be perceived visually when Δ*E***ab* > 3 [21]. It can be seen from Figure 1G that red wine with microwave treatment at 1 d resulted in Δ*E***ab* of 13.6, 16.7, and 20.2 for 100 W, 200 W, 300 W, respectively. These Δ*E***ab* values were perceptible to the human eye, but all the treated wine presented Δ*E***ab* values lower than 3 from 10 d to 60 d, indicating Δ*E***ab* that was imperceptible to the naked eye. However, all treated wine presented Δ*E***ab* higher than 3 at 70 d, indicating that Δ*E***ab* of microwave-treated red wine become more evident with the extension of storage time, and the Δ*E***ab* of red wine treated at 100 W was most apparent. As can be observed in Figure 1H, red wine treated at 30 °C presented Δ*E***ab* lower than 3, indicating that Δ*E***ab* was not perceptible to the human eye; that is to say, treatment at low temperature had little effect on Δ*E***ab*. Δ*E***ab* was obvious in red wine under 40 °C and 50 °C at 1 d, then Δ*E***ab* became less obvious from 10 d to 60 d, with Δ*E***ab* of 14.7 and 17.5 for 40 °C and 50 °C, respectively, at 70 d. According to *L**, *a**, *b**, Δ*E***ab* results, it was concluded that microwave power of 100 W and microwave temperature of 40 °C or 50 °C should be used throughout the experiment.

### 3.2. Visible Spectrum Analysis of Microwave Treatment Time and Storage Time on Wine

The visible spectrum of red wine samples (M_0_, M_t3_ and M_t6_) stored for 1 d, 10 d, 20 d, 30 d, 40 d, 50 d, 60 d, and 70 d were analyzed. The curve profiles of microwave-treated red wine (M_t3_ and M_t6_) and untreated red wine (M_0_) were similar, as shown in Figure 2A. The absorbance curve of red wine at 380–800 nm rose with the increase of microwave treatment time, which suggested that microwave treatment could promote the formation of pigment compounds in wine. The absorbance (M_0_) of red wine at 70 d was higher than that at 1 d in the visible band, indicating that red wine also formed pigment compounds during natural aging. It should be emphasized that microwave treatment increased absorbance across the whole visible spectrum. In addition, the spectral curve scans of M_0_, M_t3_, and M_t6_ were very similar with the extension of storage time from 1 d to 70 d. As shown in Figure 2B, the curve of M_t6_ was typically profiled; the longer the storage time from 1 d to 70 d, the higher the absorbance of the visible spectrum. In the aging process, free anthocyanins participated in condensation reactions to form new oligomeric and polymeric pigments, thus changing the colour of the wine [24]; colourless non-anthocyanin phenols (including flavan-3-ols and flavonols) could form yellow pigments through multi-oxidation and decarboxylation reactions [25], which could also change the colour of the wine. Microwave irradiation could induce the production of free radicals and further initiate chemical reaction in red wine [9]. These results showed that microwave treatment induced the production of free radicals and triggered chemical reactions to accelerate the production of some stable pigments in red wine. In other words, microwave irradiation could cause change of red wine colour by accelerating the reaction speed, to reduce traditional aging time.

### 3.3. Browning Index, Wine Colour and Colour Intensity

During aging, a browning index (BI, A420) was employed to express the degree of browning caused by non-enzymatic oxidation, which was associated with oxidationreaction and polymerization of phenolic compounds in wine [26]. The pale yellow of red wine changed to intense yellow with the increase in storage time, so BI (Table 1) in all red wine samples increased gradually. The red wines treated by microwave (M_t3_ and M_t6_) were significantly higher in BI than M_0_, and the longer the microwave treatment time, the more significant the increase of BI. This phenomenon showed that microwave treatment could significantly increase red wine colour, which may be due to ionic conduction and dipole rotation under microwave irradiation. These two phenomena co-occur in most cases [7], which could induce free radicals and complex chemical reactions to form more coloured pigments. That is to say, the increase of BI under microwave treatment reflected that microwave irradiation could stimulate certain chemical reactions and induce oxidation and polymerization of phenolic compounds to form brown compounds at 420 nm.

Wine colour (WC, A520) proved to be a strictly “quantitative” parameter because it was related to the absolute pigment amounts rather than their relative percentages [27]. According to the data shown in Table 1, the red wine treated by microwave was significantly higher in colour than M_0_, and the longer the time of microwave treatment, the greater the increase of WC during storage, similar to the changing trend of BI during wine storage. Due to the gradual accumulation of coloured pigments in red wine (M_0_, M_t3_, and M_t6_), WC increased with the extension of storage time, increasing the visible spectrum at 520 nm [1]. In addition, the increase of WC in microwave-treated wine was higher, which may be because the microwave-induced free radicals were the most important trigger for chemical reactions related to red wine colouring at 520 nm [9].

The sum of the absorbance at 420, 520, and 620 nm was recorded as colour intensity (CI), the absorbances at 420 and 520 nm were related to the absorption of some pigments in type A and type B vitisins, and the absorbance at 620 nm was associated with purple hue [28]. After 70 d aging, red wine suffered an absorbance increase at 420, 520, and 620 nm, so CI of red wine increased during natural aging, and the same phenomenon was observed under microwave irradiation for 3 min and 6 min (Table 1). These results showed that the generation of new pigments made the colour of red wine more stable during aging. CI of microwave-aged red wine was higher than that of natural aging, and longer duration of microwave treatment had apparent effects on pigment generation. These results showed that longer microwave treatment could induce more free radicals and promote the formation of stable pigments, thus enhancing the colour intensity of red wine.

### 3.4. CIELab Coordinates

Table 1 shows *L**, *a**, and *b** evolution during aging in the CIELab space. In general, lightness (*L**) and *a** decreased during aging [20], and the results of *L** and *a** in all wine samples showed a decreasing trend during storage. The results in Table 1 indicate that *L** and *a** of red wine after microwave irradiation were higher than those of M_0_ with extended storage time, indicating that microwave technology could give red wine higher clarity and more redness. M_t3_ had higher *L** and *a**, indicating that short-duration microwave treatment could form more stable pigment complexes and anthocyanin derivatives. Long-duration microwave treatment could induce the possible degradation of active compounds [17], which was probably why *L** and *a** of M_t6_ were lower than those of M_t3_. Levels of *b** in M_t3_ and M_t6_ were higher than iin M_0_ during storage, and this phenomenon showed that microwave treatment could accelerate the yellowness of red wine.

*C***ab* represented vividness, which was related to *a** and *b** [29]. The change trend of *C***ab* was similar to *a** and *b** in red wine (Table 1); the larger the value of *C***ab*, the more vivid the wine. *C***ab* of M_0_ tended to decrease along with the storage, indicating that the colour of the wine was flat, and no apparent differences in *C***ab* value were observed between M_t6_ and M_0_. The *C***ab* value of M_t3_ was higher than M_0_, although there were some fluctuations of vividness. The experimental results showed that short-duration microwave treatment could improve the *C***ab* value of red wine.

All wine samples were located in the first quadrant (positive *a** and *b**), corresponding to the red region of the first quadrant (*hab* = 10~20°) [21]. During aging, the *hab* of all red wine samples tended to decrease, indicating that the colour of red wine changed from red-orange hues to more red hues; this phenomenon may be related to the formation of new red pigments [30]. In addition, the *hab* of M_t3_ demonstrated little change, which indicated that short-duration microwave treatment had strong stability effects on *hab* in red wine.

Δ*E***ab* was calculated between two colour points in the three-dimensional space defined by *L**, *a**, and *b** in the CIELab space [31]. Δ*E***ab* value of M_t3_ was higher than M_t6_, which indicated that the colour difference caused by short-duration microwave treatment was distinguishable by the naked eye. The colour difference of M_t3_ was significant (Δ*E***ab* > 6) at 70 d, indicating that the effect of the microwave treatment became more obvious as aging time extended.

### 3.5. Analysis of TPC and TMA in Red Wine

TPC and TMA of red wines treated with microwaves are shown in Figure 3A,B; the TPC and TMA contents in all wine samples gradually declined during storage. TPC and TMA decline with time in natural aging [32]. The decrease in TPC content may be due to the fact that phenolic compounds could react with electrophilic quinones to form dimers or polymers, which may rearrange their structures to form new o-diphenols, and these o-diphenols may be oxidized for further polymerization, resulting in the decrease of TPC content in wine [32]. These reactions could be favoured by microwave irradiation, so TPC content in treated red wines was lower than that of M_0_ on the treatment day. The TPC contents in M_t3_ and M_t6_ were higher than in M_0_ during storage, which might be due to more free radicals induced by microwaves leading to broken hydrogen bonds of dimers or polymers. TPC plays a significant role in certain organoleptic properties of red wine and is related to colour intensity [33]. The number of phenol groups was measured by Folin–Ciocalteu colorimetry and was observed to increase during storage, so higher TPC content could effectively improve red wine quality. the results shown in Figure 3A help to indicate that higher quality red wine could be obtained under microwave irradiation. The wine colour was modified during wine storage, the monomeric anthocyanins polymerized or reacted with other compounds to generate more stable pigments in aged wines [34], and microwave treatment had little effect on the content of TMA compared with natural aged red wine.

### 3.6. Identification and Quantification of Phenolic Compounds

The samples of M_0_, M_t3_, and M_t6_ were analyzed by the HPLC system at 280 nm. Based on the retention time and UV-vis spectrum of the corresponding standards, the main phenolic compounds, including gallic acid, caffeic acid, syringic acid, (+)-catechin, Cy-3-glu, and Mv-3-glu were separated and analyzed as shown in Figure 4.

### 3.7. Changes of the Phenolic Compounds during Storage

As shown in Figure 3C–H, the concentrations of the main phenolic compounds in red wines were calculated every ten days according to the calibration curve of corresponding standards. In all samples, the content of gallic acid was the highest, and gallic acid was the main compound in all quantitative phenolic compounds. Although there were some fluctuations in the contents during storage, the gallic acid content increased slightly within the 70 d, which may be because gallic acid was the constituent unit of polyphenols, and some polymerized phenolic compounds were hydrolyzed during the red wine storage, resulting in increased gallic acid content [1]. In short, the trends of change were consistent with the increase in gallic acid observed during storage [35]. The concentration of gallic acid had a positive correlation with astringency, and astringency played a critical role in the sensory characteristics of wine.

Caffeic acid, as a critical copigment compound, has a strong copigmentation effect. During wine aging, caffeic acid can combine with anthocyanins to form anthocyanin complexes, which have more stable structures, to improve the level of chromaticity [15]. Caffeic acid decreases with aging, and this result seems to be consistent with aged Sherry wines [35]. In addition, the data of our study reflected the higher content of stable anthocyanin complexes in red wine and a higher level of chromaticity, which was consistent with higher WC and CI in Table 1.

Syringic acid is an effective copigment, a degradation product of Mv-3-glu complexes in red wine. During storage, the degradation of Mv-3-glu complexes is inevitably accompanied by the accumulation of syringic acid [36], and the content of syringic acid gradually increases during natural aging [1]. During storage, the contents of syringic acid in microwave-treated red wine were higher than in M_0_, and this result showed that microwave treatment could promote the degradation of Mv-3-glu complexes; that is to say, microwave technology could promote the aging of red wine.

Furthermore, (+)-catechin, an important flavan-3-ol in red wine as a copigment of anthocyanins, plays a vital role in the stabilization of wine colour [37]. The reduction of (+)-catechin in all wine samples might be related to its copigmentation effect with anthocyanins, and may be responsible for stabilizing colour; these stable pigment complexes could explain the increase of WC and CI as mentioned above. Also, (+)-catechin could link with the bridge group to produce xanthylium cations [38], and the yellow oxidation products could explain the increase of BI as mentioned above. As shown in Figure 3F, the content of (+)-catechin in microwave-treated red wine was lower than that of M_0_, and this result showed that microwave treatment could increase WC, CI, and BI of red wine. In other words, microwaves could promote red wine aging.

The main anthocyanins in red wine were Cy-3-glu and Mv-3-glu, and their corresponding evolutions were very similar in Figure 3G,H. Specifically speaking, the concentrations of Cy-3-glu and Mv-3-glu in M_0_, M_t3_, and M_t6_ showed an overall decreasing trend over the storage process. In general, a decreasing trend of monomeric Cy-3-glu and Mv-3-glu during aging in red wine was observed [39]. As a result of the oxidation, self-degradation, precipitation, and formation of stable anthocyanin complexes, the concentrations of Cy-3-glu and Mv-3-glu decreased during aging [40]. Furthermore, microwave irradiation did diminsh the content of Cy-3-glu and Mv-3-glu during wine storage, and the longer the treatment time, the quicker the decline rate. In addition, the relatively faster decline rate indicated that the chemical reactions might be accelerated by appropriate microwave treatment time, resulting in quicker formation of wine colour similar to naturally aged wine.

### 3.8. Principle Component Analysis

In this study, principal component analysis (PCA) was used to illustrate the correlation pattern between the colour characteristics and the main phenolic compounds in red wine. PCA is often used to interpret data, and correlation loading plots are used to represent the results. The correlation loading plots are bounded by the positive and negative correlations of research objects. Generally speaking, closer potential variables or factors have greater positive correlation; more distant potential variables or factors have greater negative correlation. Figure 5 shows the correlation loading plots for colour characteristics and main phenolic compounds in the M_0_, M_t3_, and M_t6_ samples. The correlation patterns of the treated wines (M_t3_ and M_t6_) were similar to that of M_0_, which was consistent with the changing trend of colour characteristics and the main phenolic compounds during storage, described in Section 3.3 and Section 3.4. Specifically, in all samples these variables or factors (BI, WC, CI, *L**, *b**, *hab*, TPC, TMA, (+)-catechin, Cy-3-glu, Mv-3-glu) had higher loading on PC1, reflected in the information of the indexes of 11 variables or factors. Among these, *L**, *b**, *hab*, TPC, TMA, (+)-catechin, Cy-3-glu, and Mv-3-glu had positive correlations, while BI, WC, and CI had negative correlations. Significant positive correlations existed among *L**, *b**, *hab*, TPC, TMA and (+)-catechin, Cy-3-glu, Mv-3-glu, while significant negative correlations existed among BI, WC, CI and *L**, *b**, *hab*, TPC, TMA, (+)-catechin, Cy-3-glu, Mv-3-glu. The variables or factors of caffeic acid, Cy-3-glu, and Mv-3-glu in all samples had higher loading on PC2, reflected in the information of the indexes of 3 variables or factors, and these variables or factors had positive correlations. In general, microwave irradiation did not change the correlations between the contents of main phenolic compounds and colour characteristics. A positive correlation between TMA and *L**, *b**, *hab* was also observed [41]. A positive correlation between *b**, *hab*, and (+)-catechin, Cy-3-glu, Mv-3-glu was also observed [42], and the decrease of *b** and *hab* values might be due to the formation of sufficient pigments (anthocyanin complexes or anthocyanin derivatives), which resulted in the decline of (+)-catechin and Cy-3-glu, Mv-3-glu. Considering the increase of WC, CI and the decrease of (+)-catechin, Cy-3-glu, and Mv-3-glu during storage, the results of PCA suggested that the increase of red colour and the depth of wine might be due to the copigmentation among (+)-catechin, Cy-3-glu, and Mv-3-glu during wine storage, thus stable anthocyanin-catechin polymers were formed, and the visible spectrum was increased. Considering the increase of BI and content changes of (+)-catechin during storage, the PCA results suggested that the increase of yellow colour might be related to the formation of yellow xanthylium cation pigments through multi-oxidation and decarboxylation reactions of (+)-catechin [25]. These results showed that microwave radiation did not change the correlations between colour characteristics and main phenolic compounds, but accelerated the aging trend of colour properties and main phenolic compounds during storage.

## 4. Conclusions

According to the results of this study, the microwave irradiation of the young Carbernet Gernischt dry red wine changed its colour properties and visual spectrum, and these changes could be observed in microwave-treated red wine during storage. These results indicated that proper microwave irradiation (lower power and temperature, appropriate time) could accelerate some aging reactions in red wine that proceed slowly in untreated wine. In summary, microwave irradiation had a long-term effect on the evolution of wine colour characteristics, and phenolic compounds had a similar trend of change during storage. As we all know, oak barrel aging can bring abundant benefits and improve the quality of wine. Therefore, the combination of oak barrels and microwave technology may be more conducive to wine aging. The changing phenomena in other different varieties of red wine under microwave irradiation have not yet been studied. The changes of colour characteristics and main phenolic compounds in young Cabernet Gernischt dry red wine remain worth studying, and the mechanisms of rapid changing phenomena should be further investigating, to produce wine of similar quality to the traditionally aged product in a shorter time using microwave technology.

## Figures and Tables

**Figure 1 foods-11-01778-f001:**
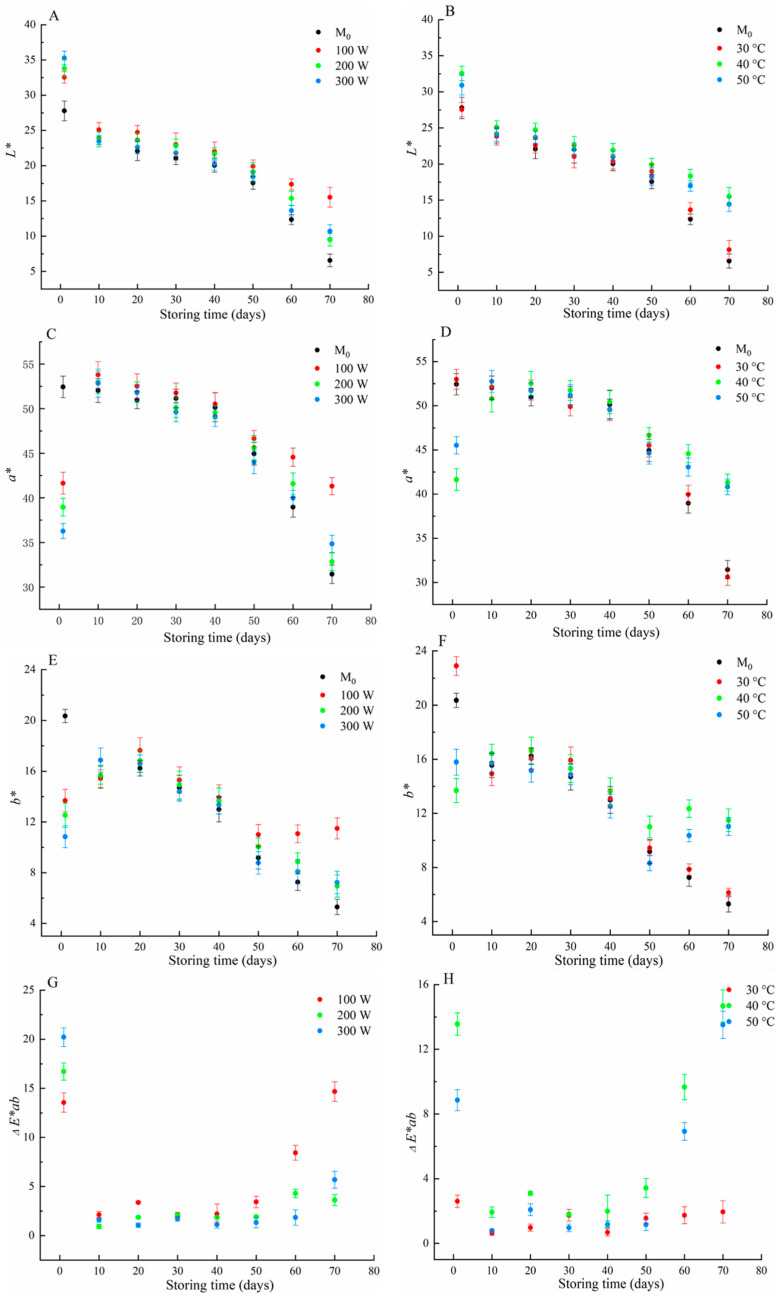
Effect of microwave power and temperature on *L**, *a**, *b**, and Δ*E***ab* of red wine during storage. (**A**) *L** value at different microwave power; (**B**) *L** value at different microwave temperature; (**C**) *a** value at different microwave power; (**D**) *a** value at different microwave temperature; (**E**) *b** value at different microwave power; (**F**) *b** value at different microwave temperature; (**G**) Δ*E***ab* value at different microwave power; (**H**) Δ*E***ab* value at different microwave temperature.

**Figure 2 foods-11-01778-f002:**
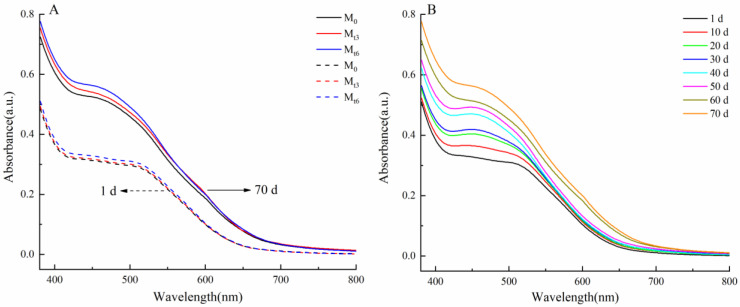
(**A**) Visible spectra of red wine samples on the first day and 70th day of storage. (**B**) Visible spectra of M_t6_ sample on the first, 10th, 20th, 30th, 40th, 50th, 60th, and 70th days of storage.

**Figure 3 foods-11-01778-f003:**
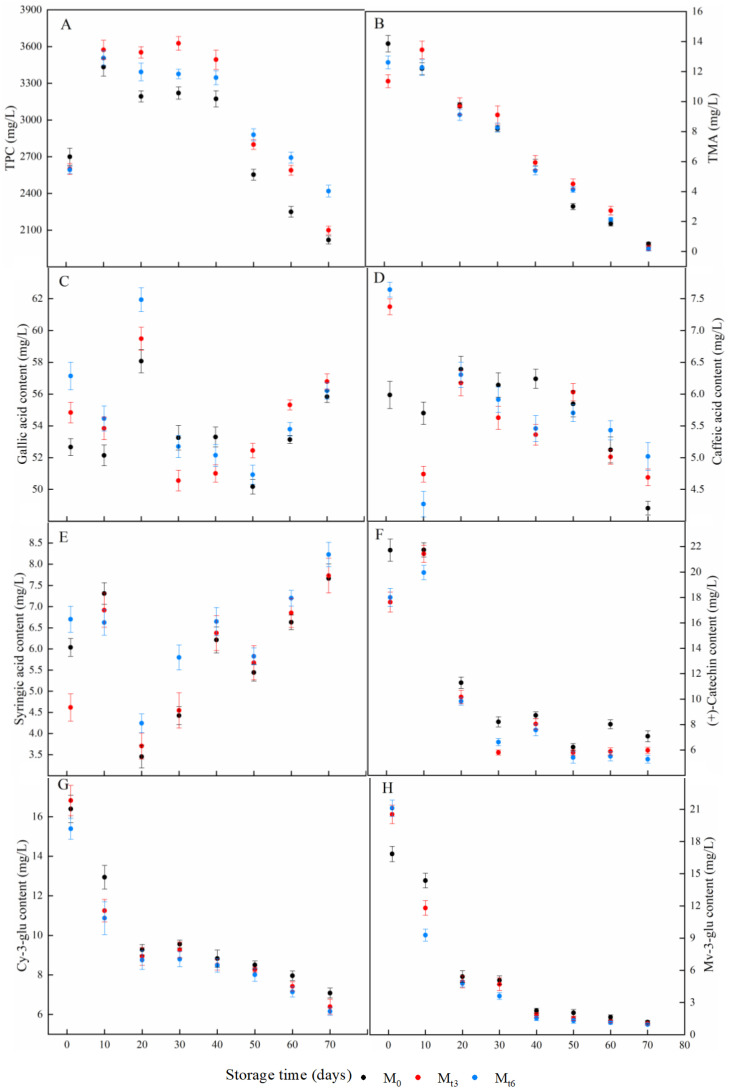
The content changes of phenolics in M_0_, M_t3_, and M_t6_ during storage. (**A**) TPC; (**B**) TMA; (**C**) gallic acid; (**D**) caffeic acid; (**E**) syringic acid; (**F**) (+)-catechin; (**G**) Cy-3-glu; (**H**) Mv-3-glu.

**Figure 4 foods-11-01778-f004:**
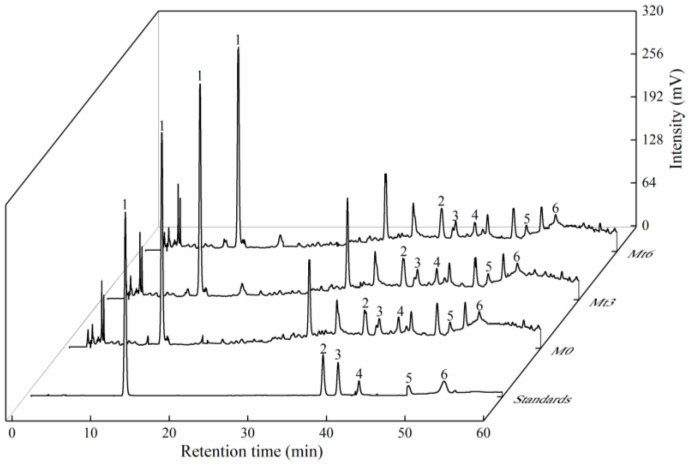
HPLC fingerprint chromatograms of standards, M_0_, M_t3_, and M_t6_ at 280 nm. 1—gallic acid; 2—caffeic acid; 3—syringic acid; 4—(+)-catechin; 5—Cy-3-glu; 6—Mv-3-glu.

**Figure 5 foods-11-01778-f005:**
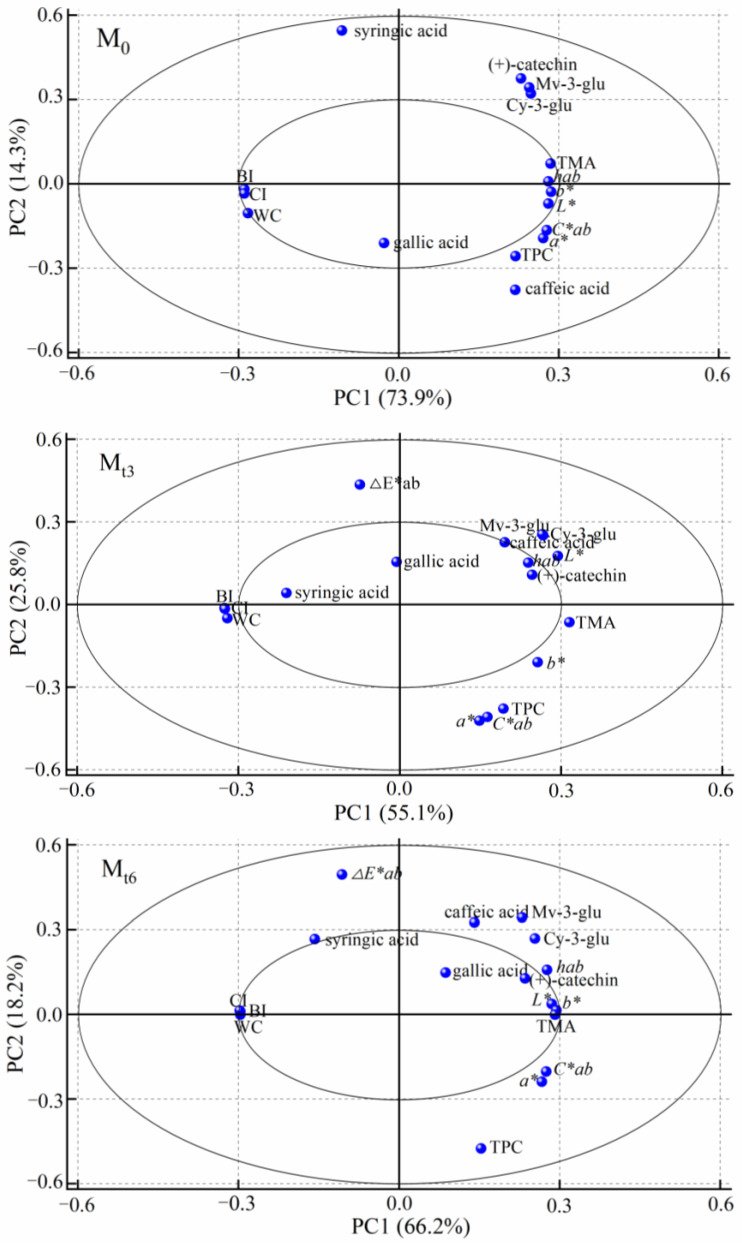
Scatter plot of PCA loading for M_0_, M_t3_, and M_t6_ during storage.

**Table 1 foods-11-01778-t001:** Effect of microwave treatment time and storage time on colour characteristics of red wine.

Parameters	Treatment Time	Storing Time(Days)							
		1	10	20	30	40	50	60	70
BI (abs 420)	M_0_	0.321 ± 0.005 ^A,a^	0.359 ± 0.004 ^A,b^	0.382 ± 0.004 ^A,c^	0.403 ± 0.007 ^A,d^	0.442 ± 0.006 ^A,e^	0.481 ± 0.007 ^A,f^	0.514 ± 0.005 ^A,g^	0.546 ± 0.006 ^A,h^
	M_t3_	0.325 ± 0.005 ^A,a^	0.362 ± 0.005 ^A,b^	0.389 ± 0.005 ^A,c^	0.411 ± 0.006 ^A,B,d^	0.458 ± 0.008 ^B,e^	0.486 ± 0.005 ^A,f^	0.526 ± 0.006 ^B,g^	0.567 ± 0.006 ^B,h^
	M_t6_	0.339 ± 0.004 ^B,a^	0.367 ± 0.006 ^A,b^	0.399 ± 0.004 ^B,c^	0.415 ± 0.004 ^B,d^	0.466 ± 0.005 ^B,e^	0.491 ± 0.004 ^A,f^	0.534 ± 0.005 ^B,g^	0.585 ± 0.005 ^C,h^
WC (abs 520)	M_0_	0.280 ± 0.008 ^A,a^	0.314 ± 0.005 ^A,b^	0.339 ± 0.004 ^A,c^	0.344 ± 0.005 ^A,c,d^	0.355 ± 0.005 ^A,d,e^	0.365 ± 0.008 ^A,e^	0.390 ± 0.110 ^A,f^	0.412 ± 0.005 ^A,g^
	M_t3_	0.284 ± 0.005 ^A,B,a^	0.317 ± 0.006 ^A,b^	0.336 ± 0.006 ^A,c^	0.348 ± 0.007 ^A,d^	0.360 ± 0.004 ^A,d^	0.378 ± 0.007 ^A,B,e^	0.401 ± 0.007 ^A,B,f^	0.428 ± 0.006 ^B,g^
	M_t6_	0.294 ± 0.005 ^B,a^	0.319 ± 0.005 ^A,b^	0.337 ± 0.007 ^A,c^	0.344 ± 0.006 ^A,c^	0.362 ± 0.007 ^A,d^	0.381 ± 0.006 ^B,e^	0.408 ± 0.006 ^B,f^	0.441 ± 0.007 ^C,g^
CI (sum of 420, 520, 620 abs)	M_0_	0.663 ± 0.007 ^A,a^	0.749 ± 0.007 ^A,b^	0.799 ± 0.007 ^A,c^	0.822 ± 0.009 ^A,d^	0.882 ± 0.006 ^A,e^	0.950 ± 0.008 ^A,f^	1.015 ± 0.008 ^A,g^	1.071 ± 0.010 ^A,h^
	M_t3_	0.672 ± 0.006 ^A,a^	0.756 ± 0.009 ^A,b^	0.801 ± 0.006 ^A,c^	0.839 ± 0.006 ^B,d^	0.910 ± 0.010 ^B,e^	0.959 ± 0.005 ^A,B,f^	1.039 ± 0.005 ^B,g^	1.130 ± 0.006 ^B,h^
	M_t6_	0.698 ± 0.010 ^B,a^	0.760 ± 0.006 ^A,b^	0.815 ± 0.010 ^A,c^	0.841 ± 0.004 ^B,d^	0.912 ± 0.008 ^B,e^	0.965 ± 0.007 ^B,f^	1.046 ± 0.010 ^B,g^	1.149 ± 0.009 ^C,h^
*L** (lightness)	M_0_	27.79 ± 0.13 ^A,h^	23.87 ± 0.22 ^A,g^	22.68 ± 0.18 ^A,f^	21.95 ± 0.11 ^A,e^	20.05 ± 0.10 ^A,d^	18.55 ± 0.13 ^A,c^	15.65 ± 0.15 ^A,b^	6.58 ± 0.17 ^A,a^
	M_t3_	32.55 ± 0.14 ^C,g^	25.10 ± 0.16 ^B,f^	22.74 ± 0.13 ^A,e^	22.02 ± 0.18 ^A,d^	21.02 ± 0.12 ^C,c^	20.91 ± 0.20 ^C,c^	18.33 ± 0.10 ^C,b^	15.33 ± 0.15 ^C,a^
	M_t6_	29.12 ± 0.17 ^B,g^	24.12 ± 0.14 ^A,f^	22.72 ± 0.21 ^A,e^	21.90 ± 0.16 ^A,d^	20.41 ± 0.19 ^B,c^	20.31 ± 0.10 ^B,c^	17.84 ± 0.18 ^B,b^	9.04 ± 0.19 ^B,a^
*a** (green/red component)	M_0_	52.43 ± 0.23 ^C,g^	52.41 ± 0.17 ^A,g^	51.82 ± 0.27 ^A,f^	51.12 ± 0.20 ^B,e^	50.15 ± 0.26 ^A,d^	44.89 ± 0.16 ^B,c^	40.65 ± 0.18 ^A,b^	31.45 ± 0.15 ^A,a^
	M_t3_	41.62 ± 0.18 ^A,a^	53.80 ± 0.22 ^B,f^	52.52 ± 0.19 ^B,e^	52.77 ± 0.18 ^C,e^	51.41 ± 0.23 ^B,d^	46.64 ± 0.20 ^C,c^	43.51 ± 0.25 ^C,b^	41.31 ± 0.21 ^C,a^
	M_t6_	49.40 ± 0.26 ^B,d^	52.50 ± 0.16 ^A,h^	51.70 ± 0.15 ^A,g^	50.62 ± 0.23 ^A,f^	50.08 ± 0.17 ^A,e^	44.22 ± 0.25 ^A,c^	41.67 ± 0.16 ^B,b^	33.30 ± 0.21 ^B,a^
*b** (yellow/blue component)	M_0_	20.35 ± 0.13 ^C,g^	15.56 ± 0.21 ^A,f^	16.23 ± 0.20 ^B,f^	14.73 ± 0.22 ^A,e^	12.97 ± 0.25 ^A,d^	9.18 ± 0.19 ^A,c^	7.63 ± 0.08 ^A,b^	5.33 ± 0.13 ^A,a^
	M_t3_	13.68 ± 0.17 ^A,c^	15.41 ± 0.17 ^A,d^	16.67 ± 0.14 ^C,e^	15.32 ± 0.085 ^B,d^	13.64 ± 0.12 ^B,c^	11.01 ± 0.16 ^C,a^	11.26 ± 0.11 ^B,a,b^	11.47 ± 0.16 ^C,b^
	M_t6_	18.35 ± 0.14 ^B,g^	15.68 ± 0.11 ^A,f^	15.02 ± 0.12 ^A,e^	14.99 ± 0.14 ^A,e^	13.13 ± 0.15 ^A,d^	10.69 ± 0.095 ^B,b^	11.09 ± 0.17 ^B,c^	6.51 ± 0.125 ^B,a^
*C***ab* (chroma)	M_0_	56.24 ± 0.26 ^C,g^	54.67 ± 0.22 ^A,f^	54.30 ± 0.20 ^B,f^	53.20 ± 0.25 ^A,e^	51.80 ± 0.31 ^A,d^	45.82 ± 0.19 ^A,c^	41.36 ± 0.19 ^A,b^	31.90 ± 0.17 ^A,a^
	M_t3_	43.81 ± 0.22 ^A,b^	55.97 ± 0.16 ^B,g^	55.10 ± 0.14 ^C,f^	54.95 ± 0.20 ^B,f^	53.19 ± 0.25 ^B,e^	47.92 ± 0.23 ^B,d^	44.94 ± 0.27 ^C,c^	42.88 ± 0.24 ^C,a^
	M_t6_	52.70 ± 0.29 ^B,e^	54.79 ± 0.12 ^A,g^	53.84 ± 0.11 ^A,f^	52.80 ± 0.26 ^A,e^	51.77 ± 0.20 ^A,d^	45.49 ± 0.26 ^A,c^	43.12 ± 0.20 ^B,b^	33.93 ± 0.23 ^B,a^
*hab* (hue)	M_0_	21.21° ± 0.05 ^C,h^	16.54° ± 0.16 ^A,f^	17.40° ± 0.29 ^B,g^	16.07° ± 0.16 ^A,e^	14.50° ± 0.19 ^A,d^	11.55° ± 0.19 ^A,c^	10.65° ± 0.06 ^A,b^	9.61° ± 0.20 ^A,a^
	M_t3_	18.19° ± 0.13 ^A,g^	15.98° ± 0.24 ^A,e^	17.61° ± 0.19 ^B,f^	16.19° ± 0.03 ^A,e^	14.84° ± 0.06 ^B,c^	13.28° ± 0.11 ^B,b^	14.52° ± 0.05 ^B,a^	15.52° ± 0.13 ^C,d^
	M_t6_	20.37° ± 0.03 ^B,f^	16.65° ± 0.16 ^B,e^	16.21° ± 0.16 ^A,d^	16.51° ± 0.08 ^B,e^	14.68° ± 0.11 ^A,B,c^	13.58° ± 0.03 ^C,b^	14.90° ± 0.16 ^C,c^	11.05° ± 0.14 ^B,a^
Δ*E***ab*	M_t3_	13.56 ± 0.56 ^e^	1.92 ± 0.065 ^b^	0.96 ± 0.20 ^a^	1.78 ± 0.46 ^b^	1.79 ± 0.36 ^b^	3.50 ± 0.15 ^c^	5.35 ± 0.23 ^d^	14.55 ± 0.31 ^f^
	M_t6_	3.87 ± 0.045 ^f^	0.38 ± 0.095 ^a^	1.26 ± 0.061 ^c^	0.61 ± 0.053 ^b^	0.42 ± 0.10 ^a^	2.42 ± 0.06 ^d^	4.22 ± 0.053 ^g^	3.30 ± 0.025 ^e^

Different lowercase letters (^a–h^) in the same row indicate significant differences (*p* < 0.05) at different storage times. Different capital letters (^A–C^) in the same column indicate significant differences (*p* < 0.05) between M_0_, M_t3_, and M_t6_.

## Data Availability

No new data were created or analyzed in this study. Data sharing in not applicable to this article.

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
