# Peer review of "Microwave Irradiation: Effects on the Change of Colour Characteristics and Main Phenolic Compounds of Cabernet Gernischt Dry Red Wine during Storage"

_foods, 2022, doi:10.3390/foods11121778_

Round 1

Reviewer 1 Report

The manuscript has an interesting topic that has practical relevance, as well. The specific research motivations are well defined. Introduction section summarizes the relevance of the study. Applied methods are adequate to the sample characteristics. Materials and methods are given clearly. The manuscript contains interesting results that are discussed with relevant references.

Comments, suggestions:

Abstract is too general and does not summarized the ’essence’ of the study.

Please check the typos and grammatical errors in the whole manuscript (see in line 30-31 ’ the producers needed to ensure the to adapt to the consumer market’, ’Carbernet’ in line 90, …for instance).

Please give the frequency of microwave treatment, as well.

Has the microbial state of wine samples tested/controlled? Has the pH of samples controlled? Change of pH has effect on the wine colour.

Authors used just one wine (Cabernet Gernischet) for the experiments. Therefore, it should better to give Cabernet Gernischet in the title instead of the general term of ’red wine’.

Figure 1 and Figure 3has low visibility.

Calibration equations in line 397-399 are unnecessary, in my opinion.

Author Response

The following is our response to the reviewer1

1) Abstract is too general and does not summarized the ’essence’ of the study.

Answer: The abstract has been modified in the revised manuscript.

2) Please check the typos and grammatical errors in the whole manuscript (see in line 30-31 ’ the producers needed to ensure the to adapt to the consumer market’, ’Carbernet’ in line 90, …for instance).

Answer: The relevant content has been revised, and the other typos and grammatical errors have also been modified in the revised manuscript.

3) Please give the frequency of microwave treatment, as well.

Answer: The frequency microwave synthesis is 2450MHz, and 2450MHz has been added in the revised manuscript.

4) Has the microbial state of wine samples tested/controlled? Has the pH of samples controlled? Change of pH has effect on the wine colour.

Answer: The microbial state of wine samples has not tested in this manuscript, but the common pathogenic bacterias (Escherichia coli, Listeria monocytogenes, Bacillus subtilis, Penicillium, Rhizopus, and Aspergillus niger) living in grape maceration stage were analyzed under microwave irradiation, the activities of the pathogenic bacterias were affected under different microwave conditions. This part of the research needs further study. Some of the results are as follows:

 Table 1 Effect of different microwave conditions on fungus inhibition rate (%)

microwave conditions

Penicillin

Rhizopus

Aspergillus Niger

100 W

19.38

58.14

1.56

300 W

27.13

61.62

14.64

500 W

37.32

71.22

28.57

700 W

46.46

76.54

59.16

1 min

24.17

41.56

6.34

3 min

27.13

61.62

14.64

5 min

28.92

74.49

24.72

Figure 1. Effect of different microwave conditions on bacterial inhibition rate (%)

pH hardly changed under microwave irradiation, the relevant research has been published[1]. Therefore, the colour of red wine was disconnected with the change of pH under microwave.

[1] Yuan, J. F., Wang, T. T., Chen, Z. Y., Wang, D. H., Gong, M. G., & Li, P. Y. (2020). Microwave irradiation: Impacts on physicochemical proper ties of red wine. CyTA-Journal of Food, 18(1), 281-290. https://doi.org/10.1080/19476337.2020.1746834.

5) Authors used just one wine (Cabernet Gernischet) for the experiments. Therefore, it should better to give Cabernet Gernischet in the title instead of the general term of ’red wine’.

Answer: Cabernet Gernischet has been added in Title.

6) Figure 1 and Figure 3 has low visibility.

Answer: Figure 1 and Figure 3 have been changed in colour with the form of point scatter.

7) Calibration equations in line 397-399 are unnecessary, in my opinion.

Answer: The calibration equations of six phenolic compounds were deleted in revised manuscript.

Reviewer 2 Report

Dear Authors,

The article raises an interesting issue of the influence of the physical method on the organoleptic properties of a food product. In my opinion, the manuscript needs to be corrected. Detailed comments below:

1) Chapter 2.2 - give the microwave frequency (not everyone remembers that the wavelength is inversely proportional to its frequency, pay attention to the phenomenon of dispersion)

2) Chapter 2.4 - CIELAB is your main research tool - describe in more detail its essence (methodology)

2a) CIELAB is described in: DOI: 10.3390 / su12187473

3) Chapter 2.8 - You applied Waller-Duncan's post-hoc test which indicates you were using the parametric ANOVA version; note that applying a parametric test requires checking the initial conditions:

- normality of the data population distribution - provide the type of test used and its result,

- homogeneity of variance in the samples - provide the type of test used and its result,

- was the minimum size of the research sample determined (?), was the assumed sample size sufficient for the tests assumed in the methodology (?), was the sample size sufficient for ANOVA (?),

4) Fig. 1 - the figure is not legible, the data in the graph should not be combined (this suggests data prediction), a better solution is to use the trend line (the least squares method), leave the data in the graph in the form of point scatter

5) Fig. 3 - as above + consider drawing in color

6) Tab. 1 - is extensive, requires correction in accordance with the MDPI format, consider including it as a material in the supplement

7) Chapter 3. Results and discussion - divide the chapters into "Results" and "Discussion" (optionally insert the sub-chapter "Discussion" into chapter 3) - at the decision of the Authors

8) Chapter 4. Conclusions - this chapter has the form of a summary, redact it into constructive conclusions, indicate application applications (or / and change the title to "Summary")

Author Response

The following is our response to the reviewer2

1) Chapter 2.2 - give the microwave frequency (not everyone remembers that the wavelength is inversely proportional to its frequency, pay attention to the phenomenon of dispersion)

Answer: The frequency microwave synthesis is 2450MHz, and 2450MHz has been added in the revised manuscript.

2) Chapter 2.4 - CIELAB is your main research tool - describe in more detail its essence (methodology). CIELAB is described in: DOI:10.3390/su121874731)

Answer: The parameters of CIELAB are described in methodology.

3) Chapter 2.8 - You applied Waller-Duncan's post-hoc test which indicates you were using the parametric ANOVA version; note that applying a parametric test requires checking the initial conditions:

- normality of the data population distribution - provide the type of test used and its result,

Answer: The type of test is Shapiro-Wilk, and p>0.05, the results show normality of the data population distribution.

- homogeneity of variance in the samples - provide the type of test used and its result,

Answer: The type of test is Pearson’s chisquared and p>0.1, the results show homogeneity of variance in the samples.

- was the minimum size of the research sample determined (?), was the assumed sample size sufficient for the tests assumed in the methodology (?), was the sample size sufficient for ANOVA (?)

Answer: Three parallel samples were determined in this research. The sample size enables analysis of the results according to the results of normality and homogeneity.

4) Fig. 1 - the figure is not legible, the data in the graph should not be combined (this suggests data prediction), a better solution is to use the trend line (the least squares method), leave the data in the graph in the form of point scatter

Answer: Figure 1 has been changed in colour with the form of point scatter.

5) Fig. 3 - as above + consider drawing in color

Answer: Figure 3 has been changed in colour with the form of point scatter.

6) Tab. 1 - is extensive, requires correction in accordance with the MDPI format, consider including it as a material in the supplement

Answer: The MDPI format of Tab. 1 has been provided in the revised manuscript. The content of Tab. 1 is related to colour parameters at different treatment time and storage time, which is more important, it is better to put it in the main manuscript.

7) Chapter 3. Results and discussion - divide the chapters into "Results" and "Discussion" (optionally insert the sub-chapter "Discussion" into chapter 3) - at the decision of the Authors

Answer: Results and discussion together also could clarify well the phenomenon and probable reasons of colour change, so put “results and discussion” together in the revised manuscript.

8) Chapter 4. Conclusions - this chapter has the form of a summary, redact it into constructive conclusions, indicate application applications (or / and change the title to "Summary")

Answer: the conclusions has been minor revised in the revised manuscript.

Round 2

Reviewer 1 Report

Authors have made a significant revision on the manuscript and provided detailed answers for reviewers' questions. Rephrasing, amendment, more detailed discussion of results with additional references made the manuscript more complete. Change of the title and modifications of the MS made the manuscript clear. The overall scientific quality of the mnauscript has been improved significantly due to the revision. I agree and accept all  modifications made by the authors.